# Stem Cell Models for Breast and Colon Cancer: Experimental Approach for Drug Discovery

**DOI:** 10.3390/ijms23169223

**Published:** 2022-08-17

**Authors:** Nitin T. Telang

**Affiliations:** Cancer Prevention Research Program, Palindrome Liaisons Consultants, Montvale, NJ 07645-1559, USA; ntelang3@gmail.com

**Keywords:** breast and colon cancer, cancer stem cells, therapeutic alternatives

## Abstract

The progression of the early stages of female breast and colon cancer to metastatic disease represents a major cause of mortality in women. Multi-drug chemotherapy and/or pathway selective targeted therapy are notable for their off-target effects and are associated with spontaneous and/or acquired chemotherapy resistance and the emergence of premalignant chemo-resistant cancer-initiating stem cells. The stem cell populations are responsible for the evolution of therapy-resistant metastatic disease. These limitations emphasize an unmet need to develop reliable drug-resistant cancer stem cell models as novel experimental approaches for therapeutic alternatives in drug discovery platforms. Drug-resistant stem cell models for breast and colon cancer subtypes exhibit progressive growth in the presence of cytotoxic chemo-endocrine therapeutics. The resistant cells exhibit upregulated expressions of stem cell-selective cellular and molecular markers. Dietary phytochemicals, nutritional herbs and their constituent bioactive compounds have documented growth inhibitory efficacy for cancer stem cells. The mechanistic leads for the stem cell-targeted efficacy of naturally occurring agents validates the present experimental approaches for new drug discovery as therapeutic alternatives for therapy-resistant breast and colon cancer. The present review provides a systematic discussion of published evidence on (i) conventional/targeted therapy for breast and colon cancer, (ii) cellular and molecular characterization of stem cell models and (iii) validation of the stem cell models as an experimental approach for novel drug discovery of therapeutic alternatives for therapy-resistant cancers.

## 1. Introduction

Despite advances in conventional and/or targeted therapy, female breast and colon cancer represent major causes of mortality in women. The American Cancer Society estimated the projected combined incidence of newly diagnosed breast and colon cancers as 333,230 and the number of cancer-related deaths as 68,060 in 2022 [1]. Multi-drug conventional chemotherapy for breast and colon cancer includes the use of several mechanistically distinct cytotoxic pharmacological agents. In contrast, molecularly targeted pathway-selective therapeutics include small molecule inhibitors such as selective estrogen receptor modulators/degraders, cyclin-dependent kinase inhibitors, aromatase inhibitors, non-steroidal anti-inflammatory drugs, selective cyclooxygenase inhibitors and selective ornithine decarboxylase inhibitors for the two organ site cancers [2].

Long-term therapeutic interventions using these pharmacological agents are notable for their off-target effects. In addition, conventional/targeted chemo-endocrine therapy is associated with spontaneous and/or acquired therapy resistance and the emergence of chemo-resistant cancer-initiating premalignant stem cell populations that are at risk for developing metastatic lesions that may be susceptible to natural products [3]. These limitations inherent to current chemo-endocrine therapy emphasize the need to develop reliable stem cell models for breast and colon cancers and to identify efficacious pharmacological agents and natural products that inhibit cancer stem cells.

It is interesting to note that estrogens affect the carcinogenic process in the breast, as well as in the colon. The role of estrogens in breast cancer is well-documented. The natural estrogen estradiol 17-β functions as a natural ligand to estrogen receptors α and β that represent ligand-regulated transcription factors. Genomic signal transduction involves a complex cascade of events culminating with estrogen response element-mediated target gene expression. Estrogen receptor-α (ER-α) is noted for its growth promoting effects [4]. In contrast, ER-β-mediated signal transduction involves a distinct transcriptional activity, leading to the negative regulation of growth in endocrine responsive target cells [5]. However, the role of estrogens in colon cancer is less defined. Published evidence suggests that ERs function as inhibitory modifiers for colon cancer. In the Apc ^MIN^/+ model for genetically predisposed colon cancer, a lack of ER-α and ER-β accelerates colon carcinogenesis [6]. ER-β functions as a negative growth regulator in colon cancer, and several naturally occurring phyto-estrogens function as potent ER-β agonists [7,8]. Collectively, this evidence suggests that ER-β agonists may function as effective negative growth regulators of breast and colon cancers.

Natural products including dietary phytochemicals, micro-nutrients, nutritional herbs and their constituent bioactive compounds represent relatively nontoxic agents. In traditional Chinese medicine, herbal formulations, prepared mostly from nutritional herbs, have been used to treat cancer patients [9,10,11]. These natural products, because of their documented long-term human consumption and preclinical growth inhibitory efficacy, may represent testable therapeutic alternatives in treating therapy-resistant breast and colon cancers.

Drug-resistant cancer-initiating stem cells evolve into metastatic phenotypes through multi-step cascades involving the activation of survival pathways and epithelial–mesenchymal transition [12,13,14]. These aspects of cancer stem cell biology represent a scientifically robust rationale for the drug discovery of novel agents effective against cancer stem cells.

The main objective of the present review is to discuss published evidence relevant to (i) conventional/targeted therapy for breast and colon cancers and their limitations, (ii) cancer stem cell biology and the development of breast and colon cancer stem cell models and (iii) the validation of stem cell-based experimental approaches as a platform for novel drug discovery for therapy-resistant breast and colon cancers.

## 2. Experimental Models

### 2.1. Parental Breast and Colon Cancer Cell Lines

Breast and colon carcinoma-derived established cell lines provide valuable cellular models for clinical cancer subtypes. Human female breast carcinoma-derived cell lines differ in the expression status of hormone receptors and growth factor receptors, and represent valuable cellular models for luminal A, luminal B, HER-2-enriched and triple-negative breast cancer subtypes. Human colon carcinoma-derived cell lines include the HCA-7 model (wild type APC, wild type, β-catenin), the SW480 model (mutant APC, wild type β-catenin) and the HCT-116 model (wild type APC, mutant β-catenin). It is notable that only the HCA-7 model was derived from a female patient, while the other two models were derived from male patients. Colonic epithelial cell lines have been established from female mice that express germline mutations in clinically relevant Apc and DNA mismatch genes [15,16,17,18]. These cell lines exhibit hyper-proliferation, accelerated cell-cycle progression, downregulated apoptosis anchorage, independent growth in vitro and tumor formation in vivo. The cell lines carrying the Apc gene mutation exhibit chromosomal instability and predominantly the aneuploid phenotype, while those carrying DNA mismatch repair gene mutations exhibit micro-satellite instability and predominantly the diploid phenotype [19,20,21].

The characteristics of parental cell lines are described in Table 1. These cell lines are developed for clinically relevant luminal A and triple-negative breast cancer subtypes and for genetically predisposed familial adenomatous polyposis (FAP) and hereditary non-polyposis colon cancer (HNPCC) subtypes. It is notable that these aberrantly hyper-proliferative cell lines exhibit AI colony formation in vitro and tumor development in in vivo. AI colony formation represents a specific and sensitive in vitro surrogate end point for tumorigenic transformation. Experimental modulation of this end point facilitates the quantitation of risk for cancer development.

### 2.2. Conventional/Targeted Therapy

Clinically relevant conventional and targeted therapeutic options are described in Table 2. These pharmacological agents are notable for their off-target effects, spontaneous/acquired therapy resistance and the emergence of drug-resistant cancer stem cell population [22,23]. Table 2 provides examples of mainstream therapeutic options for breast cancer subtypes such as luminal A, luminal B and triple-negative breast cancer (TNBC).

Table 3 exemplifies mainstream treatment options for colon cancer that include genetically predisposed familial adenomatous polyposis (FAP), hereditary non-polyposis colon cancer (HNPCC) and sporadic colon cancer subtypes.

### 2.3. Pharmacological Inhibition

The susceptibility of the cell lines to prototypic chemo-endocrine therapeutics facilitates the selection of appropriate experimental models. The cells treated with the therapeutic agents at their respective maximal cytostatic (IC_90_) concentrations inhibit anchorage independent colony formation. The selective estrogen receptor modulator tamoxifen (TAM) represents the endocrine treatment of choice for the luminal A breast cancer subtype. Treatment of the MCF-7 model for the luminal A subtype with TAM reduces the number of AI colonies. Cytotoxic chemotherapy with doxorubicin (DOX) represents the treatment of choice for the triple-negative subtype. Treatment of the MDA-MB-231 model for the triple-negative subtype with DOX reduces the number of AI colonies.

The inhibitory effects of pharmacological agents for colon cancer are illustrated in Figure 1A,B. The pharmacological agents for the FAP model include difluro-methyl ornithine (DFMO), celecoxib (CLX), 5-fluoro-uracil (5-FU) and sulindac (SUL). These agents at their respective maximal cytostatic (IC_90_) concentrations lead to a significant reduction in AI colony number. The pharmacological agents for the HNPCC model include SUL, DFMO and 5-FU. These agents at their respective maximally cytostatic (IC_90_) concentrations lead to a significant inhibition of the number of AI colonies. It is notable that the rank order of inhibitory efficacy of the pharmacological agents for AI colony formation in the FAP model is DFMO > CLX > 5-FU = SUL, and for the HNPCC model it is SUL = DFMO = 5-FU. These data in rank order suggest that the mechanisms responsible for growth inhibitory response may differ in two models.

These experiments compare the data from the multiple treatment groups to a common control group. The statistical analyses of these data are performed by analysis of variance (ANOVA) and Dunnett’s multiple comparison post-hoc test with a threshold of α = 0.05, using the Microsoft Excel 2013 XLSTAT-Base software.

### 2.4. Drug-Resistant Stem Cell Models

In stem cell biology, the stem cell population plays important roles in normal and cancerous growth. In normal epithelial organ sites, stem cells regulate cell proliferation and apoptosis for cellular homeostasis and tissue regeneration. Cell signaling pathways such as Wnt/β-catenin, Notch and Hedgehog are responsible for the maintenance of the normal stem cell population [24]. Cancer stem cells represent a minor subpopulation of chemo-endocrine therapy-resistant cancer-initiating premalignant cells intrinsic to the primary cancer. In cancer stem cells, normal regulatory pathways are disrupted, and cancer cell survival pathways are activated. These survival pathways facilitate signaling via RAS/BRAF/ MEK/ERK, PI3K/AKT and mTOR pathways, resulting in a growth advantage to the cancer cell phenotype and effective drug resistance [13,22,25].

The common and unique characteristics of normal and cancer stem cells have been utilized to develop reliable stem cell models to understand normal stem cell biology and to facilitate mechanism-based investigations focused on drug discovery for testable alternatives for therapy-resistant cancer. For example, stem cell signaling pathways such as Wnt, Hedgehog NOTCH and p38 MAPK [26,27], and cell surface markers such as CD44, CD133 and ALDH1 [28,29] may represent therapeutic targets for pharmacological small molecule inhibitors as well as for naturally occurring compounds.

In addition to the stem cell models for breast and colon, similar models for other organ site cancer are developed. For example, the models for head and neck cancer utilize stem cell-enriched tumor spheroids [30]. Stem cell models for pancreas utilize pluripotent stem cells [31,32]. These stem cell models provide valuable experimental approaches to investigate phenotypic pluripotency, plasticity, therapy-resistant tumor progression and to identify new compounds that may be effective in targeting stem cell population [33,34].

For the isolation of chemo-resistant breast and colon cancer stem cells, growth resistance to prototypic chemo-endocrine therapeutics is used to select putative stem cell phenotypes.

The selection of putative drug-resistant stem cells for breast and colon cancer models is described in Table 4.

Long-term maintenance of the cells at high concentrations of TAM and DOX for breast cancer and SUL and 5-FU for colon cancer results in the elimination of drug-sensitive phenotypes, as evidenced by the decreased viable cell number, and the progressive growth of the resistant phenotypes, as evidenced by the increased number of tumor spheroids [35,36]. Actively growing putative stem cells are expanded in the presence of cytotoxic concentrations of TAM or DOX. The putative stem cells exhibit an increased number of tumor spheroids (TS) in the TAM-R model for Luminal A breast cancer subtype, and in DOX-R model for TNBC subtype. The data presented in Figure 2A–D illustrate the effective isolation of putative stem cells in the breast cancer models. The statistical significance of the data for viable cell number is determined by the two-sample Student’s *t* test, and that for tumor spheroid number is determined by the Chi square (X^2^) test.

The putative stem cells exhibit an increased number of tumor spheroids in SUL-R for the FAP subtype and in the 5-FU-R model for the HNPCC subtype. The data presented in Figure 3A–D illustrate the effective isolation of putative stem cells in the colon cancer models. The statistical significance of the data on viable cell number is analyzed by the two-sample Student’s *t* test, and that for tumor spheroid number is analyzed by the X^2^ test.

The characterization of drug-resistant stem cells is accomplished by determining the status of select stem cell-specific cellular and molecular markers. These markers included the cell surface proteins CD44 and CD133, and nuclear transcription factors NANOG, OCT-4 and c-Myc. Several nuclear transcription factors such as OCT-4, Klf-4, SOX-2 and c-Myc also represent essential factors for the survival of induced pluripotent stem cells [38,39]. These molecular endpoints are quantified by cellular uptake of relevant FITC conjugated antibodies, and the data are expressed as log mean relative fluorescent units per 10^4^ fluorescent events [40]. The data summarized in Table 5 demonstrate that the drug-resistant phenotypes exhibit upregulated expressions of the molecular markers for cancer stem cells.

## 3. Experimental Modulation

Evidence from integrative oncology has documented that herbal formulations used in traditional Chinese medicine represent effective interventions for the prevention/treatment of cancer [9,10,11]. Recent evidence has also suggested that natural products such as dietary phytochemicals [41,42], natural products [43] and Chinese nutritional herbs, as well as their constituent bioactive compounds [44], function as negative growth regulators via targeting cancer stem cell signaling pathways to overcome chemo-resistance.

### 3.1. HER-2-Enriched Breast Cancer

For the treatment of the hormone receptor positive/HER-2 positive luminal B and hormone receptor negative/HER positive HER-2-enriched breast cancer subtypes, the small molecule inhibitor of EGF and /HER-2 receptors lapatanib is commonly used. The 184-B5/HER cell line represents a cellular model for the hormone receptor-negative HER-2-positive HER-2-enriched breast cancer subtype. The 184-B5/HER cells provided the lapatinib-resistant (LAP-R) stem cell model that was developed by the selection and expansion of the surviving cell population in the presence of LAP. The data in Table 6 illustrate that the naturally occurring terpene carnosol effectively inhibits tumor spheroid formation and downregulates the expression of CD44, NANOG and OCT-4. The statistical significance of the data for TS are determined by the X^2^ test, and that for CD44, NANOG and OCT-4 are determined by the two-sample Student’s *t* test.

### 3.2. FAP Model for Colon Cancer

The non-steroidal anti-inflammatory drug sulindac (SUL) has documented preclinical efficacy in the 850^MIN^/+ mouse model for FAP syndrome. SUL is also used in the clinical therapy of FAP patients and patients with sporadic colon cancer.

The 1638N COL cells provided the sulindac-resistant (SUL-R) stem cell model that was developed by the selection and expansion of surviving cell population in the presence of SUL. The data in Table 7 illustrate that curcumin (CUR), a bioactive agent present in the Asian spice turmeric, inhibits TS formation and downregulates the expressions of CD44, CD133 and c-Myc. The statistical significance of the data for TS are determined by the X^2^ test, and that for CD44, CD133 and c-Myc are determined by the two-sample Student’s *t* test.

## 4. Conclusions

The development of drug-resistant stem cell models for breast and colon cancer subtypes validates mechanism-driven experimental approaches that may prioritize drug discovery of novel alternatives that target drug-resistant stem cells. The data discussed in the present review provide a scientifically robust rationale for additional experiments focusing on the experimental modulation of ubiquitous transcription factors in cell signaling pathways of drug-resistant cancer stem cells [11,12,13,14,37,41,42,43,44,45,46].

## 5. Future Research

Spontaneous or acquired resistance to conventional and molecularly targeted therapy, and potential cross-resistance to individual therapeutic agents represent a formidable challenge for new drug discovery of stem cell targeting compounds [12,13,14,47,48,49].

Telomerase, an RNA–protein complex, adds hexameric repeats of 5′-TTAGGG-3′ to the telomeres and regulates DNA replication in normal proliferating cells prior to the onset of replicative senescence. Somatic cells lack the expression of telomerase. However, in immortalized cells and in cancer-derived cells, this enzyme is re-expressed in cells that escape replicative senescence [50,51]. Differentiation-inducing agents, natural products and small molecule inhibitors may represent novel testable alternatives for telomerase inhibitory activity.

Small molecule inhibitors functioning as epigenetic modifiers [33,34] may represent testable alternatives for drug resistant stem cells. Evidence has been published from organoid cultures for head and neck cancer [30], non-small cell lung cancer [52], and pancreatic ductal adenocarcinoma [31,32,53,54]. This evidence provides scientifically robust rationale for future research directions. Collectively, the mechanistic leads for telomerase inhibitors and epigenetic modifiers suggest their efficacy independent of therapy resistance in cancer stem cells.

It needs to be recognized that data generated from established cell lines are dependent on extrapolation for their clinical relevance and translatability. Future investigations focusing on the development of stem cell models from patient-derived therapy-resistant tumor samples, tumor organoids [55,56,57,58,59,60] and ex vivo organ culture explants [61,62] may reduce the need for extrapolation, and facilitate the discovery of novel efficacious compounds.

## Figures and Tables

**Figure 1 ijms-23-09223-f001:**
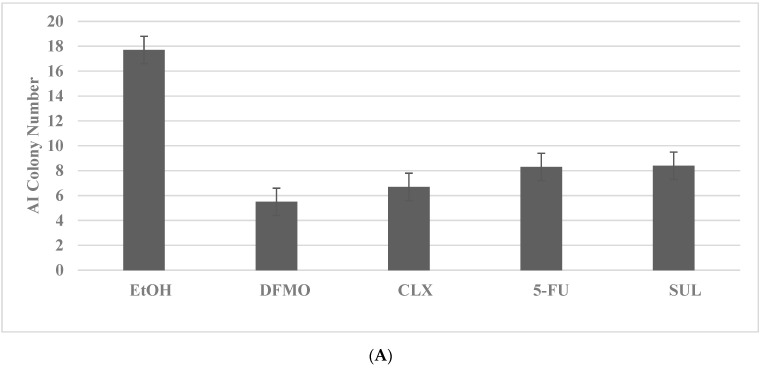
Growth inhibitory effects of pharmacological agents on colon cancer models. (**A**) Reduction in AI colony number by treatment with DFMO, CLX, 5-FU and SUL in the FAP model for colon cancer. EtOH versus DFMO and CLX *p =* 0.038; EtOH versus 5-FU and SUL *p* = 0.04. (**B**) Reduction in AI colony number by treatment with SUL, DFMO and 5-FU in the HNPCC model. EtOH versus SUL and DFMO *p =* 0.037; EtOH versus 5-FU *p =* 0.04. Data analyzed by ANOVA with Dunnett’s multiple comparison post-hoc test. AI, anchorage independent; EtOH, ethanol; DFMO, difluoro-methyl ornithine; CLX, celecoxib; 5-FU, 5-fluoro-uracil; SUL, sulindac.

**Figure 2 ijms-23-09223-f002:**
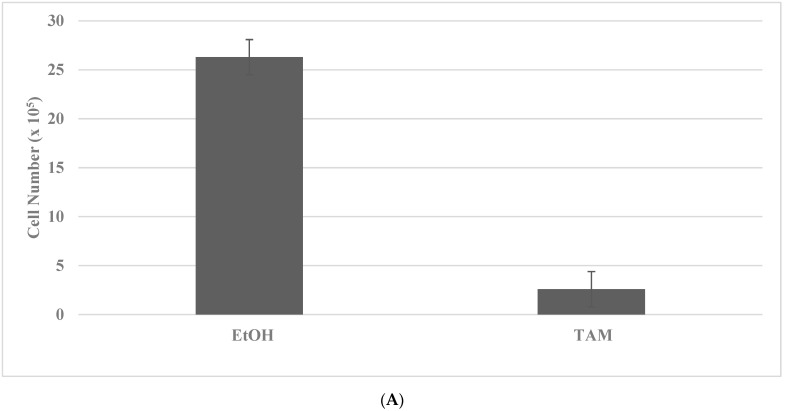
Isolation of drug-resistant putative stem cells in breast cancer models. (**A**) Decreased viable cell number by treatment with TAM in TAM-S cells. EtOH versus TAM *p =* 0.01. (**B**) Increased tumor spheroid number in TAM-R cells. TAM-S versus TAM-R *p =* 0.01. (**C**) Decreased viable cell number by treatment with DOX in DOX-S cells. EtOH versus DOX *p =* 0.01. (**D**) Increased tumor spheroid number in DOX-R cells. DOX-S versus DOX-R *p =* 0.01. Data analyzed by the two-way Student’s *t* test. TAM, tamoxifen; TAM-S, TAM sensitive; TAM-R, TAM resistant; EtOH, ethanol; DOX, doxorubicin; DOX-S, DOX sensitive; DOX-R, DOX resistant (data summarized from [35,37]).

**Figure 3 ijms-23-09223-f003:**
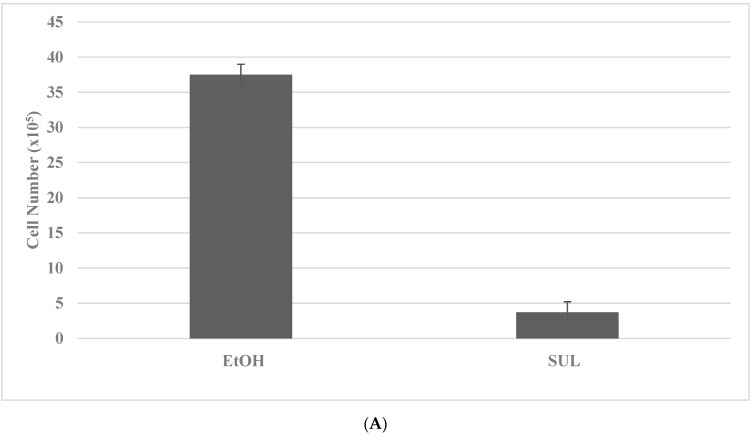
Isolation of drug-resistant putative stem cells in colon cancer models. (**A**) Deceased viable cell number by treatment with SUL in SUL-S cells. EtOH versus SUL *p =* 0.01. (**B**) Increased tumor spheroid number by treatment with SUL in SUL-R cells. SUL-S versus SUL-R *p =* 0.01. (**C**) Decreased viable cell number by treatment with 5-FU in 5-FU-S cells. EtOH versus 5-FU *p =* 0.01. (**D**) Increased tumor spheroid number by treatment in 5-FU-R cells5-FU-S versus 5-FU-R *p =* 0.01. SUL, sulindac; SUL-S, SUL sensitive; SUL-R, SUL resistant; 5-FU, 5-fluoro-uracil; 5-FU-S, 5-FU sensitive; 5-FU-R, 5-FU resistant (Data summarized from [36,37]).

**Table 1 ijms-23-09223-t001:** Cellular models for breast and colon cancer.

Parental Cells	Clinical and Biological Markers						Clinical Subtype
	ER	PR		HER-2	AICF	Tumor	
Breast							
184-B5	-	-		-	-	-	Normal breast
MCF-7	+	+		+	+	+	Luminal A
MDA-MB-231	-	-		-	+	+	Triple-negative
**Colon**	**Apc ^[+/+]^**		**Mlh1 ^[+/+]^**				
C57 COL	+		+		-	-	Normal colon
1638N COL	-		+		+	+	Apc mutation codon 1638. FAP
Mlh1/1638N COL	-		-		+	+	HNPCC

ER, estrogen receptor-α; PR, progesterone receptor; HER-2, human epidermal growth factor receptor-2; AICF, anchorage independent colony formation; Apc, adenomatous polyposis coli genotype; Mlh1, Mut-L genotype; FAP, familial adenomatous polyposis; HNPCC, hereditary non-polyposis colon cancer.; Genotype ^[+/+]^, status of alleles.

**Table 2 ijms-23-09223-t002:** Conventional chemo-endocrine therapy and targeted therapy for breast cancer.

Organ Type	Therapy		Clinical Subtype
	Conventional	Targeted	
**Breast**	Multi-drug combination DOX, PCT, CPT	SERM, CDKI, AI, HER-2 inhibitors	Luminal A, Luminal B, TNBC

DOX, doxorubicin; PCT, paclitaxel; CPT, carboplatin; SERM, selective estrogen receptor modulator; CDKI, cyclin dependent kinase inhibitor; AI, aromatase inhibitor; HER-2, human epidermal growth factor receptor-2, TNBC; triple-negative breast cancer.

**Table 3 ijms-23-09223-t003:** Conventional chemotherapy and targeted therapy for colon cancer.

Organ Type	Therapy		Clinical Subtype
	Conventional	Targeted	
**Colon**	Multi-drug combination 5-FU, OPN, ITC	NSAID COX-2 inhibitors ODC inhibitors	FAP, HNPCC sporadic cancer

5-FU, 5-fluro-uracil; OPN, oxaliplatin; ITC, irinotecan; NSAID, non-steroidal anti-inflammatory drugs; COX-2, cyclooxygenase-2; ODC, ornithine decarboxylase; FAP, familial adenomatous polyposis; HNPCC, hereditary non-polyposis colon cancer.

**Table 4 ijms-23-09223-t004:** Drug-resistant stem cell models.

Agent	Concentration	Stem Cell Model
**Breast**		
Tamoxifen (TAM)	1.5 µM	TAM-R Luminal A
Doxorubicin (DOX)	0.5 µM	DOX-R TNBC
**Colon**		
Sulindac (SUL)	10 µM	SUL-R, FAP
5-fluoro-uracil (5-FU)	0.2 µM	5-FU-R, HNPCC

TAM-R, tamoxifen resistant; DOX-R, doxorubicin resistant; TNBC, triple-negative breast cancer; SUL-R, sulindac resistant; FAP, familial adenomatous polyposis; 5-FU-R, 5-fluoro-uracil resistant; HNPCC, hereditary non-polyposis colon cancer (Data summarized from [35,36]).

**Table 5 ijms-23-09223-t005:** Marker expression in breast and colon cancer stem cell models.

Organ Type	Molecular Marker			Clinical Subtype
	CD44	NANOG	OCT-4	
**Breast**				
TAM-R	+	+	+	Luminal A
DOX-R	+	+	+	TNBC
**Colon**	**CD44**	**CD133**	**c-Myc**	
SUL-R	+	+	+	FAP
5-FU-R	+	+	+	HNPCC

Expression of CD, cluster of differentiation; NANOG, DNA binding homeobox transcription factor; OCT-4, octamer binding transcription factor-4; c-Myc, cellular Myc. Data expressed as log mean RFU and + symbol denotes upregulated expression relative to the drug-sensitive phenotype. TAM-R, tamoxifen-resistant; DOX-R, doxorubicin-resistant; SUL-R, sulindac-resistant; 5-FU-R, 5-fluoro-uracil-resistant; RFU, relative fluorescent unit (Data summarized from [37]).

**Table 6 ijms-23-09223-t006:** Experimental modulation of stem cell markers in the lapatinib-resistant (LAP-R) breast cancer model.

Treatment	Concentration	Stem Cell Markers			
		TS	CD44	NANOG	OCT-4
DMSO	0.1%	14.8 ± 1.9	20.8 ± 4.4	11.8 ± 3.1	14.2 ± 3.8
CSOL	5 µM	1.9 ± 0.2	2.9 ± 0.6	2.7 ± 0.7	2.9 ± 0.9
X^2^		7.74			
*p*-value		0.010	0.010	0.020	0.020
Inhibition		87.2%	86.0%	77.1%	79.6%

TS, tumor spheroid number; CD44, cluster of differentiation 44; NANOG, DNA binding homeobox transcription factor; OCT-4, octamer binding transcription factor-4. Data expressed as log mean RFU. DMSO, dimethyl sulfoxide; CSOL, carnosol; RFU, relative fluorescent unit (Data summarized from [37]).

**Table 7 ijms-23-09223-t007:** Experimental modulation of stem cell markers in the Sulindac-resistant SUL-R colon cancer model.

Treatment	Concentration	Stem Cell Markers			
		TS	CD44	CD133	c-Myc
EtOH	0.01%	20.1 ± 4.0	15.9 ± 3.4	16.7 ± 3.4	8.3 ± 1.8
CUR	10 µM	4.0 ± 0.3	3.5 ± 0.3	5.3 ± 0.5	3.8 ± 0.4
X^2^		7.74			
*p*-value		<0.01	0.01	0.01	0.03
Inhibition		80.1%	77.9%	68.3%	54.2%

TS, tumor spheroid number; CD, clusters of differentiation; c-Myc, cellular Myc. Data expressed as RFU. EtOH, ethanol; CUR, curcumin; RFU, relative fluorescent unit (Data summarized from [36]).

## Data Availability

The data sets used in this review are available from the author on reasonable request.

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
