# Peer review of "Stem Cell Models for Breast and Colon Cancer: Experimental Approach for Drug Discovery"

_ijms, 2022, doi:10.3390/ijms23169223_

Round 1

Reviewer 1 Report

It is an unusual review article that includes many experimental data in the manuscript. The increased CSC activity in drug resistance breast or colon cancer is well-known, and the author should present the literature review results more systematically. Major comments are listed.

1. The conventional therapies for breast or colon cancer depend on their molecular types. Table 2 should be divided according to their molecular types, not mixed them up.

2. What is the” AI colonies” represented in Figure 1? Please clearly describe it.

3. Several grammatical issues should be corrected, such as “m TOR” in line 170 should be corrected as “mTOR”; there should be a comma between “Hedgehog” and “NOTCH” in line 176. Whole manuscript should be checked carfully.

4. It is not clear if the data presented in Table 2, Figure 2, Figure 3, and Table 3 to 6 were obtained from the author himself. If yes, a “Materials and Methods” section should be added. If not, please add the references.

5. In Table 5, it is not clear about the meanings of the provided numbers, especially for Stem cell markers. Please provide the unit information in the footnote. The same issue also applies to Table 6.

6. The author should provide more discussions in the “Future Research” section. For example, any suggestion for manipulating CSC activity by any agent or strategy to overcome drug resistance.

Author Response

Many thanks for constructive critique and helpful recommendations from the two reviewers. I have considered all the recommendations and have made appropriate revisions in the originally submitted manuscript. Newly added segments in the text of the revised version are highlighted in bold face. The following is my point-by-point response to individual comments.

Reviewer 1:

  1. Concern: Table 2 should be revised based on the molecular subtypes for breast and colon.

Response: The original Table 2 has now been presented as Table 2 for breast subtypes and Table 3 for colon subtypes. The text has been revised accordingly.

  1. Concern: Figure 1: AI colonies

Response: Anchorage independent colonies are abbreviated as AI colonies. This term has been described in detail in the text.

  1. Concern: The grammatical mistakes should be corrected.

Response: The grammatical mistakes in the entire text of the revised manuscript have been checked and corrected.

  1. Concern: Clarify the origin of the data in Table 2, Figures 2 and 3, and Tables 3-6.

Response: The data is summarized from published primary data. The original publications contain detailed materials and method section. In the present manuscript the summarized format for Tables and Figures is used to illustrate and discuss the relevant evidence. The relevant references are included in the footnotes of individual Tables and Figures.

  1. Concern: Tables 5 and 6 should be revised to include details about the stem cell markers.

Response: The technical details of quantitative parameters for stem cell markers are provided in the foot notes of the two tables. The text is revised to include a summary of methodology.

  1. Concern: The Future research section should be expanded to include discussion regarding experimental manipulation of cancer stem cell activity.

Response: This section is revised to discuss limitations of spontaneous/acquired drug resistance that hamper stem cell targeted new drug discovery and potentially effective strategies to overcome drug resistance. This section contains recently published evidence that provides mechanistic leads about the efficacy of epigenetic modifiers that affect stem cell pluripotency in cancer organoid models developed from induced pluripotent stem cells. Potential significance of these and other agents as stem cell targeting new drugs is also addressed. 

Reviewer 2 Report

The author presented a well-described review article describing cancer stem cell models for the drug discovery using breast and colon stem cells. The article is well-written, and the author has many publications relevant to this topic. Apart from self- cited (relevant) references, this article has some minor mistakes which need to be addressed before making any decision on this paper.

1.    Author needs to explain a little about the role of other cancer stem cells apart from breast and colon for the drug discovery. If you cannot include that, please update the title because it is confusing and states that you cover all cancer stem cell models in your article.

2.    Line 143-148 is in bold. Is this by mistake, or does the author wants to give stress some important points?

3. Figures 1A and 1B were prepared from the author’s published papers. The author must have the data from which they prepared the graphs. Please include the statistics and make graphs more transparent using Graphpad Prism etc.

4.     The paper is full of abbreviations without their complete forms. Please check and include it. i.e., AI, DFMO, CLX, 5-FU, SUL, etc.

5.    Line 267: - A: Deceased viable…. Was it ‘Deceased’ or ‘Decreased’?

6.    The author, at last, must mention drug resistance, which hampers the experimental approach to drug discovery.

Author Response

Reviewer 2:

  1. Concern: Systematic discussion of published evidence relevant to cancer stem cell activity and drug resistance.

Response: Collectively, discussion of cancer stem cells in the text of revised sections of this manuscript (Introduction: paragraph 5, Drug-resistant stem cell models: paragraph 1 and revised text, and Future Research: revised text), provides an updated systematic discussion of published evidence on drug-resistant cancer stem cells.

   Note that updated Reference section contains a total of 62 references that include 43 references (about 69%) that are published within last 5 years.   The self-cited references are essential to discuss the relevant published evidence. These self-cited references represent only about 11% of the total number of references.       

  1. Concern: Role of other cancer stem cells.

Response: There is evidence in the literature about the cancer stem cell models developed for organ sites other than breast and colon. For example, cancer stem cell models for non-small cell lung cancer, pancreas, and prostate have been developed. These models involve the use of induced pluripotent stem cells and cancer organoids.  These models are applicable for new drug discovery of stem cell targeting agents. These aspects are discussed in the sections on drug-resistant stem cells and future research, respectively. (See also the response to comment # 6 from Reviewer 1.  

   The title of the manuscript has been revised to specify breast and colon cancer stem cell models.   

  1. Concern: Lines 143-148 are in bold face.

Response: In response to the preliminary recommendations from the editorial office these lines were included in bold face. The content of these lines should have been corrected by the editorial office prior to sending the manuscript to the reviewers.

  1. Concern: Figure 1A and 1B should include the statistics,

Response: The statistical significance of unpublished data presented in Figures 1A and 1B is determined by comparing the treatment arms (pharmacological agents) with the common solvent (ethanol) control arm. The statistical analysis is performed using ANOVA with Dunnett’s multiple comparisons test. These details are included in the text and calculated P values are included in the footnotes.     

  1. Concern: The abbreviations should be expanded.

Response: The abbreviations are expanded at their first use in the text and in the footnotes of the Tables and Figures.

  1. Concern: Line 267.

Response: The term “Deceased” is corrected to Decreased.

  1. Concern: Limitation of drug resistance in the experimental approach to drug discovery.

Response: Spontaneous/acquired drug resistance, together with pluripotency, epithelial-mesenchymal transition and telomerase re-expression represent hallmarks of cancer stem cells. These properties of stem cells represent major technical limitations in new drug discovery of stem cell targeting agents. This aspect is addressed in the revised Future Research section, together with potential mechanistic leads for the efficacy of telomerase inhibitors and epigenetic modifiers. These agents may represent testable therapeutic alternatives that function independent of drug resistance in drug-resistant stem cell population.    

Round 2

Reviewer 1 Report

Although the authors have done a great improvement on this manuscript, Table 2 and Table 3 still require to be revised to arrange the therapeutics used in each molecular type of breast or colon cancer by setting different columns, not mixing them up in one column.

Author Response

Reviewer 1:

Concern: Revise Table 1 and Table 2.

Response: Pursuant to the recommendation by the reviewer, the two tables are revised to include the content in separate columns.

Reviewer 2:

Concern: This reviewer has not raised any concerns/issues for the Round 2 of the review.

Response: Response is not required.